∂ | **Open Peer Review** | Clinical Microbiology | Research Article

# Community-associated quinolone-resistant and extended-spectrum beta-lactamase-producing *Escherichia coli* isolates are similar to clinical infection isolates by sequence type and resistome

Emily E. Benedict,[1] Wesley Agee,[1] Tiffany Hink,[2] Katelyn L. Parrish,[2] Kimberly A. Reske,[2] Kate Peacock,[2] Rachel E. Bosserman,[2] Alyssa Valencia,[2] Akshay Saluja,[2] Elianora Ovchiyan,[2] Olivia Arter,[2] Kevin Jolani,[2] Erik R. Dubberke,[2] Gautam Dantas,[1,3,4,5,6] Jennie H. Kwon[2]

**ABSTRACT** *Escherichia coli* is a public health threat capable of causing multiple types of infection, carrying a variety of antimicrobial resistance genes (ARGs), and disseminating ARGs to other microbes. Since ARG-carrying *E. coli* can exist as a commensal gut microbe, intestinal *E. coli* in community-associated (CA) members presents an under-appreciated reservoir of ARGs. We cultured 75 CA *E. coli* isolates from stool of 64 patients lacking inpatient healthcare exposures >24 hours in the previous 12 weeks. Remnant stool submitted to the Barnes-Jewish Hospital (BJH) microbiology laboratory for *Clostridioides difficile* testing was plated to MacConkey agar with ciprofloxacin and extended-spectrum beta-lactamase (ESBL) Chrome Agar to isolate resistant *E. coli* colonies, which were whole-genome sequenced. Isolates were compared to ESBL *E. coli* genomes published by Mahmud et al. (B. Mahmud, M. A. Wallace, K. A. Reske, K. Alvarado, et al., mSystems 7:e00519-22, 2022, https://doi.org/10.1128/msystems.00519-22), which were collected from bloodstream and urinary tract infections. We identified ESBL genes and quinolone resistance elements in *E. coli* isolates from all patients, 32 (50%) of whom had no recent antibiotic exposure. Sequence type (ST) 131 isolates carried more quinolone resistance elements but fewer ESBL genes than other STs. Eleven patients carried two distinct *E. coli* lineages simultaneously. CA ESBL *E. coli* displayed a lower diversity of beta-lactamase genes but similar rates of antibiotic resistance genes compared to ESBL *E. coli* reported by Mahmud et al. (https://doi.org/10.1128/msystems.00519-22). Carriage of resistance elements without recent antimicrobial exposure suggests the presence of circulating, resistant *E. coli*. Our results show the continually evolving resistance profile of CA *E. coli*, demonstrating the importance of characterizing antimicrobial resistance in the community.

**IMPORTANCE** Antimicrobial-resistant *Escherichia coli* presents a substantial threat to public health, limiting treatment options and potentially horizontally transferring its resistance to other members of the gut microbiome. Resistance to quinolones and beta-lactams, specifically, hinders treatment of urinary tract and gastrointestinal infections, both commonly caused by *E. coli*. Tracking successful lineages, such as ST131, within the healthcare setting can inform clinicians about resistance patterns among their patients, but this work shows that other STs present an even higher antimicrobial resistance burden than ST131. In addition to monitoring multiple lineages of antimicrobial-resistant *E. coli*, it is necessary to identify and understand community-associated carriage of this organism, as evidenced by the increasing prevalence of community-associated ESBL *E. coli* carriage and our specific results showing similar resistance burdens within the clinic and community. This work presents insight into antimicrobial-resistant

**Peer Reviewer** Sadia Isfat Ara Rahman, International Centre for Diarrhoeal Disease Research, Dhaka, Bangladesh

Address correspondence to Jennie H. Kwon, jkwon@northwestern.edu, or Gautam Dantas, dantas@wustl.edu.

Emily E. Benedict and Wesley Agee contributed equally to this article. Author order was decided by roles in figure generation and revisions.

The authors declare no conflict of interest.

See the funding table on p. 13.

*E. coli* among those without significant healthcare exposures, providing important community-focused surveillance that is currently lacking.

**KEYWORDS** *Escherichia coli*, antibiotic resistance, genomics

Antimicrobial-resistant *Escherichia coli* strains are a priority pathogen for the World Health Organization (1) and an antimicrobial resistance threat cited by the U.S. Centers for Disease Control and Prevention (2). Though many *E. coli* strains persist as commensal members of a healthy gut microbiome (3, 4), several pathogenic *E. coli* strains are major causes of bloodstream infections (BSIs) and urinary tract infections (UTIs), both in the United States (5) and globally (6–8). Since 2000, *E. coli* has been highlighted as a significant public health threat, in part due to the frequent carriage of the extended-spectrum beta-lactamase (ESBL) gene $bla_{CTX-M}$, particularly the allelic variant $bla_{CTX-M-15}$ (8, 9). Sequence type (ST) 131 is successful as an extraintestinal pathogen and antimicrobial-resistant lineage, imposing a significant public health burden (10). One subclade of this ST, *H30*, was found to be the largest driver of ST131 expansion in the United States (10–12). This subclade can be further distinguished by two emerging sub-lineages: H30R1, a fluoroquinolone-resistant subset, and H30Rx, a strain defined by its co-resistance to both fluoroquinolones and beta-lactams. Importantly, ST131 has been found in the community with increasing prevalence (4, 10, 13, 14).

ESBLs are capable of hydrolyzing penicillins and third-generation cephalosporins, dramatically limiting treatment options for community-based infections (15). Classical beta-lactamases, $bla_{TEM}$ and $bla_{SHV}$, emerged in the late 20th century (16, 17), primarily in Enterobacteriaceae. Currently, $bla_{CTX-M-15}$ and $bla_{CTX-M-14}$ are the most commonly reported ESBLs in human-isolated *E. coli* (9, 15). The mortality rate of bacteremia caused by ESBL-producing *E. coli* was observed to be three times higher than for susceptible *E. coli*, demonstrating the severe health threat posed by drug-resistant *E. coli* (18). Concerningly, ESBL-producing *E. coli* are increasingly prevalent in the community setting (6, 19), particularly in the form of asymptomatic carriage (presence of the organism in the absence of a clinically significant infection), with an estimated prevalence of 21% intestinal carriage globally (19, 20). Accordingly, continued surveillance of reservoirs in the community setting is needed to combat the dynamic threat presented by ESBL-producing *E. coli*.

Fluoroquinolones are commonly prescribed in the United States for a variety of infections, including UTIs and gastrointestinal infections (21, 22). Therefore, fluoroquinolone resistance in *E. coli* has major implications for public health, both in community and healthcare settings (23). Key mutations in the quinolone resistance-determining regions of *E. coli* occur in genes encoding DNA gyrase (*gyrA*, *gyrB*) and topoisomerase IV (*parC*, *parE*), reducing the drug's binding ability and thereby diminishing its antibacterial activity (24, 25). Additionally, mutations in the AcrAB-TolC efflux system can prompt its overexpression and further reduce drug effectiveness by actively removing intracellular fluoroquinolones (26). Since these resistance mutations are predominantly in chromosomally encoded alleles, they can persist in clonal lineages and be stably inherited by daughter cells, even in the absence of antibiotic selective pressures. The stability and pervasiveness of fluoroquinolone resistance in human-associated microbial communities pose a significant threat to human health.

The continued global expansion of these resistant *E. coli* lineages in the community supports the urgent need for enhanced surveillance, antimicrobial stewardship, and novel therapeutic strategies to mitigate their impact on public health. It is essential that we understand the community carriage of *E. coli* harboring these resistance mechanisms, particularly as carriage among people in the community is increasing faster than carriage among hospitalized patients (19, 20). Historically, hospital- and healthcare-associated *E. coli* has been better monitored, and patients identified as carrying antimicrobial-resistant *E. coli* can be placed on contact precautions or given antibiotics to limit their risk of transmitting the organism (2, 27, 28). Comparatively, *E. coli* in the community is both

harder to survey and its transmission is more complicated to mitigate (20). To address this gap in surveillance, we genomically characterized a group of ciprofloxacin-resistant or ESBL-producing *E. coli* isolates from people in St. Louis, Missouri, without recent significant inpatient healthcare exposures. We also compare the antimicrobial resistance in these community-associated (CA) ESBL *E. coli* with a set of ESBL *E. coli* collected from BSIs and UTIs in St. Louis, Missouri (29).

## MATERIALS AND METHODS

### Study description

Samples were selected from a biospecimen repository of remnant stool specimens originally collected for *Clostridioides difficile* testing at Barnes-Jewish Hospital (BJH) in St. Louis, Missouri, between May 2017 and August 2021. CA diarrhea was defined as stool submitted for testing within 72 hours of hospital admission or at an outpatient office visit from patients with no inpatient healthcare exposures of ≥24 hours in the 12 weeks prior to stool collection (inpatient healthcare exposures included inpatient hospitalization or residence in a long-term care facility, nursing facility, or rehabilitation facility). The BJH Medical Informatics database was queried to determine relevant healthcare exposures, which were then verified during subsequent chart review for each patient to ensure that all network and non-network healthcare exposures were identified. Among patients with more than one stool submitted for *C. difficile* testing during this timeframe, only the first available stool per patient was included. Electronic medical records were reviewed to collect clinical data points, namely demographics, comorbidities, symptoms, healthcare exposures, infections, and medications in the 12 weeks prior to stool collection (including antibiotics, laxatives, immunosuppressants, steroids, and chemotherapy), for patients from whom *E. coli* was isolated. Patients less than 18 years of age were excluded. Abstracted data were entered into a REDCap database, and descriptive statistical analyses of this clinical data were performed using SPSS version 29.0 (IBM, Armonk, New York).

### Culturing, identification, and antimicrobial susceptibility testing of isolated *E. coli*

After storage at −80°C, PBS was added to each remnant stool before homogenization and plating to ESBL Chrome Agar (Hardy Diagnostics, Santa Maria, California), which is selective for ESBL-producing organisms, and MacConkey agar with ciprofloxacin (Hardy Diagnostics, Santa Maria, California), which is selective for quinolone-resistant organisms. Agar plates were incubated aerobically at 35°C and protected from light. Plates were checked for growth after 24 hours and 48 hours. Each distinct colony morphotype was counted and identified using VITEK MS MALDI-TOF (bioMerieux, Durham, North Carolina). All isolates identified as *E. coli*, including multiple colonies from the same selective plate, were tested for antimicrobial susceptibility using antibiotic disk diffusion according to the 31st edition of the CLSI guidelines. Isolate-level media type and relevant patient metadata can be found in Tables S1 and S2.

### DNA extraction and sequencing

All isolates identified as *E. coli* ($n = 102$) were subjected to whole-genome sequencing. First, genomic DNA was extracted using the QIAmp BIOstic Bacteremia DNA Kit (Qiagen, Germantown, Maryland, USA). Genomic DNA, at a concentration of 0.5 ng/µL as measured with the PicoGreen dsDNA assay (Thermo Fisher Scientific, Waltham, Massachusetts), was then prepared for short-read sequencing with the Nextera XT Kit before being submitted for Illumina sequencing on the NovaSeq X Plus platform, targeting 2 million reads per barcode for $2 \times 150$ bp reads.

## Read processing and genome assembly

Following short-read sequencing, raw reads were demultiplexed by index pair and quality-trimmed with Trimmomatic v0.39 (30). The resulting fastq files were filtered using FastQC v0.11.9 (31). Cleaned reads were assembled into draft genomes via *de novo* assembly with SPAdes v3.15.3 (32). Draft genome quality was determined using CheckM v1.2.1 (33); assemblies with completeness $\leq$90% or contamination $\geq$5% were considered low quality and removed from further analysis. Species identification of assemblies was performed with Mash v2.3 (34) (Mash shared hashes $\geq$ 700; Mash-provided database refseq.genomes.k21s1000.msh). A total of 94 assembled genomes from 64 patients were identified as *E. coli* and high quality (completeness $\geq$90%, contamination <5%) to be carried into further analysis. CheckM results can be found in Table S3.

## Genomic annotation and phylogenetic analysis

Multi-locus sequence typing was completed using mlst v2.22.1 (35) according to PubMLST typing schemes (36) and can be found in Table S3. Genomes were function-ally annotated with Prokka v1.14.6 (37), and the results were used to construct a core-genome alignment with Panaroo v1.5.1 (38) (core threshold 95%; strict mode). Maximum-likelihood trees were constructed in RAxML v8.2.12 (39) with 100 bootstraps and were visualized in iTOL v7 (40) and the R packages ggplot2 v3.5.2 (41) and ggtree v3.12.0 (42). Antimicrobial resistance genes (ARGs) were annotated with AMRFinderPlus v2.1.6 (43), and the results can be found in Table S4. Please note that these results are not phenotypically confirmed, although we supplemented these genomic annota-tions with the susceptibility tests described above. We used R v4.4.0 (44) and R Studio v2025.09.0+387 (45) throughout this work.

## Single-nucleotide polymorphism and network analyses

SNP-dists v8.5.0 (46) was used to identify core-genome single-nucleotide polymor-phism (SNP) distances between isolate genomes, using the multi-sequence alignment obtained from Panaroo as described above. To determine an SNP threshold for clonality, we compared all-versus-all pairwise SNP distances and conducted distribution-based clustering, identifying the number of core-genome SNPs above which there is a distinct split in the distribution. This method suggested an appropriate threshold of 22 SNPs (Fig. S1). Applying this criterion of $\leq$22 SNPs to isolate genomes recovered from the same patient stool, we were left with 75 unique isolates. Further comparisons across patient stools using this threshold were used to identify strain networks, as has been previously used in relevant literature (47–50). These SNPs were used to construct distance networks, using the qgraph v1.9.8 (51) package in R for study isolates, in which edges were undirected and edges were inversely weighted by number of SNPs between isolates. This package was also used for analysis of ESBL *E. coli* from this study combined with isolates from Mahmud et al. (29) to visualize strain modules with a directed graph and edges inversely weighted by the number of SNPs between isolates. Pairwise core-genome SNP distances can be found in Table S3.

## Study description

A published study reported on a group of ESBL *E. coli* isolates obtained from blood and urine specimens from hospitalized patients at BJH (29). All patients from the original sample set were admitted to BJH medical and/or oncology wards from June 2016 to December 2019 and had clinical ESBL *E. coli* UTI and/or BSI. The original analysis included 149 *E. coli* isolates from 129 patients. The present analysis was restricted to isolates from blood and urine specimens collected from 2018 to 2019 in order to more closely match the time period of the isolates in this study, and only one isolate per patient was included. The final specimen set included 87 ESBL *E. coli* isolates from 87 unique patients, none of whom appear in the sample set from the present study. For the 87 patients/isolates included in the Mahmud et al. sample, chart review was performed in

the electronic medical record (Epic) to collect demographics and medications in the 12 weeks prior to specimen collection (antibiotics, immunosuppressants, and chemotherapy). Data were entered into a REDCap database, and descriptive statistics of this clinical data were analyzed using SPSS version 29.0 (IBM, Armonk, New York).

## Whole-genome sequencing and *de novo* assembly

As described in the study by Mahmud et al. (29), all 87 isolates were subjected to whole-genome sequencing on the NovaSeq 6000 (Illumina) platform to obtain 2 × 150 bp reads. Resulting reads were demultiplexed by index pair, followed by quality trimming with Trimmomatic v0.38 (30) and removal of contaminating human reads using Deconseq v4.3 (52) with the hsref38 database. Clean paired reads were sub-sampled using seqtk v1.3 (53), targeting 2 million reads per isolate, and were used to generate draft genomes via *de novo* assembly with SPAdes v3.15.3 (32). Draft genome quality was determined with CheckM v1.2.1 (33); assemblies with completeness <90% or contamination ≥5% were considered low quality and removed from further analysis.

## Statistics

All summary clinical analysis was performed using SPSS v.29.0 (IBM, Armonk, New York), as described above. All non-clinical statistical analyses were performed with R and R Studio. Counts and proportions of resistance genes and elements were compared using the Wilcoxon rank-sum test. We applied a significance threshold of $P < 0.05$ throughout.

## RESULTS

### Patients and isolates in this study

From the BJH stool repository, 691 stools were identified as CA and were eligible for culture, and 102 *E. coli* isolates were recovered from 64 patients (Fig. 1a). After SNP-based dereplication, a total of 75 unique isolates were included in genomic analysis. Half of the participants ($n = 32$) reported no antibiotic exposure of any kind in the 12 weeks preceding specimen collection (Table 1). Of antibiotic-exposed patients, most did not receive antibiotics in classes relevant to our selective culture conditions, namely penicillins ($n = 8$ received), cephalosporins ($n = 11$ received), carbapenems ($n = 0$ received), and quinolones ($n = 14$ received). In total, 52 of 75 (69%) isolates in our sample grew only under ciprofloxacin selection, and 23 of 75 (31%) isolates grew only under ESBL selection, with zero isolates recovered from both selections (Fig. 1c). ST131 was the most common ST ($n = 39$, 52%), followed by ST1193 ($n = 7$, 9%), which intriguingly was only recovered during ciprofloxacin selection (Fig. 1c).

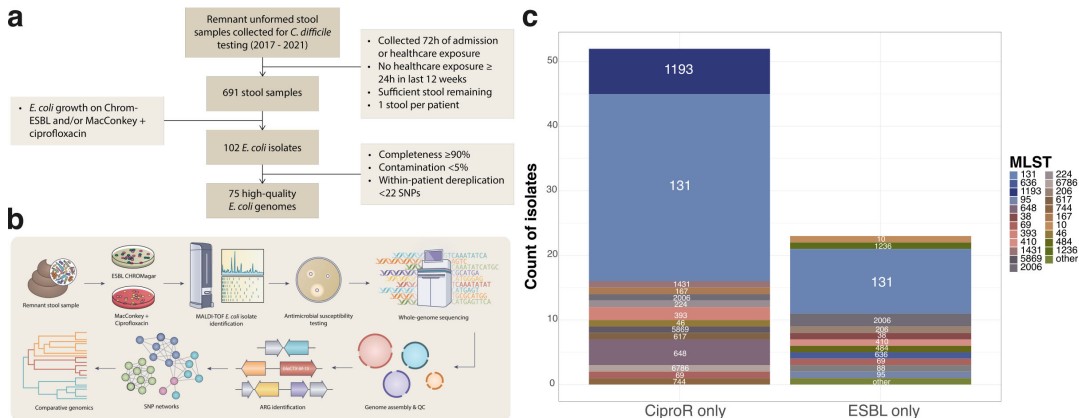

**FIG 1** Study overview and workflow. (a) Flow chart describing patient inclusion criteria and isolate inclusion criteria. (b) Schematic of study workflow. (c) Number of isolates by media type, colored and labeled by sequence type. SNP = single-nucleotide polymorphism; MLST = multi-locus sequence type; h = hours; ESBL = extended-spectrum beta-lactamase; CiproR = ciprofloxacin resistant.

**TABLE 1** Patient characteristics in this study

| Variable | Mean (range) or *N* (%) |
|---|---|
| Age, mean (range) | 56 years (18-94 years) |
| Female, *N* (%) | 31 (48) |
| Patient location at sample collection, *N* (%) | |
| Inpatient | 44 (69) |
| Outpatient | 17 (26) |
| Emergency department | 3 (5) |
| Medication in last 12 weeks, *N* (%) | |
| Any antibiotics | 32 (50) |
| Penicillins | 8 (12) |
| Cephalosporins | 11 (17) |
| Carbapenems | 0 (0) |
| Quinolones | 14 (22) |
| Immunosuppressant | 15 (23) |
| Chemotherapy | 12 (19) |

To confirm clonality among isolates cultured from the same patient stool, we compared pairwise core-genome SNPs. Distribution-based clustering identified an empirical cutoff of 22 SNPs to determine unique strain lineages (Fig. S1), which is consistent with prior work (47–50). Isolates obtained from the same patient had very similar collections of ARGs (Fig. S2), despite eight of 11 isolates being from different selective agars. Applying this criterion to our set of samples resulted in 75 unique isolate genomes from 64 patients. We observed multi-strain colonization of enteric *E. coli* in 11 of 64 (17%) included patients based on our defined SNP cutoff (Fig. 2a). Furthermore, slightly more than half (six of 11) of co-colonized patient samples shared strains of the same ST, whereas the remaining co-occurring isolates (five of 11) were of unique STs (Fig. 2b).

There were four instances of isolates fewer than 22 core-genome SNPs apart recovered from different patients in this study, involving isolates from STs 131, 2006, 1193, and 393 (Fig. 3a). Two pairs of isolates were isolated from specimens collected

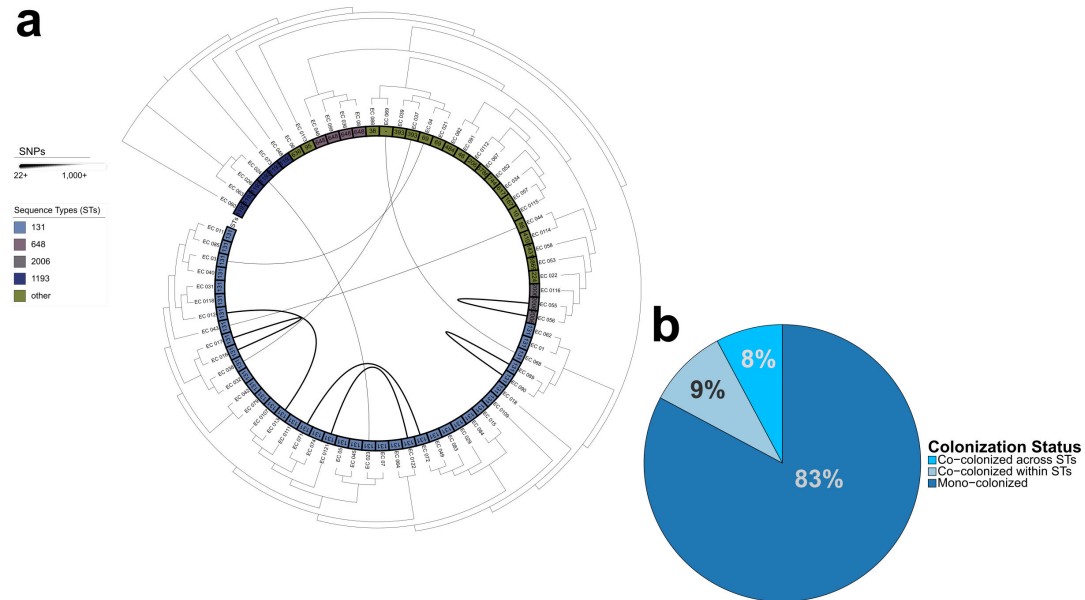

**FIG 2** Within-patient co-colonization. (a) Maximum-likelihood tree of 75 *E. coli* genomes with connecting edges that correspond to the magnitude of reported SNPs between same-patient isolates. (b) Pie chart displaying *E. coli* carriage among our patient cohort, illustrating the percentage of mono-colonized and co-colonized patient samples. SNP = single-nucleotide polymorphism; ST = sequence type.

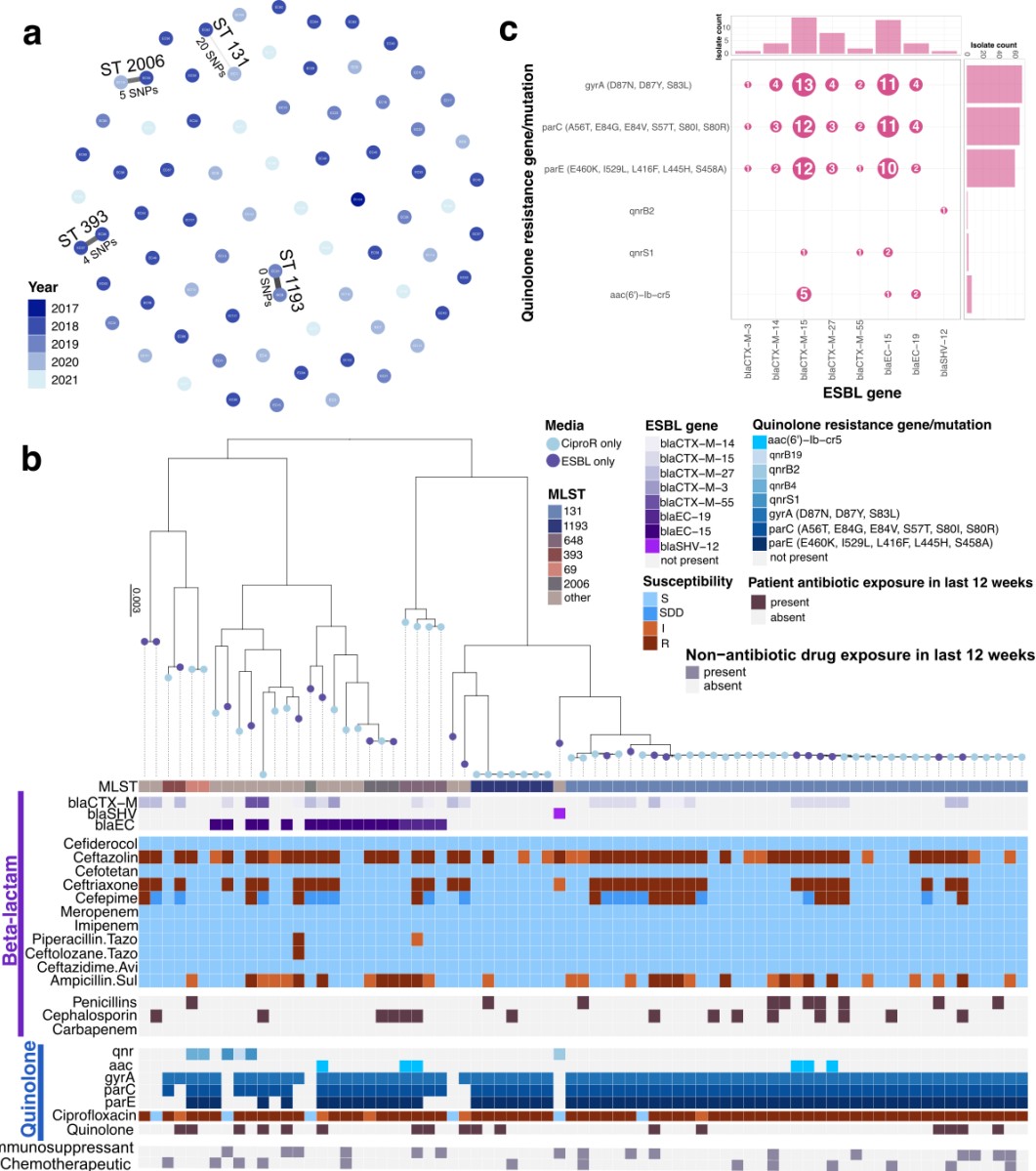

**FIG 3** Relatedness of *E. coli* and co-carriage of ESBL genes and quinolone resistance elements. (a) Network of all isolates in this study paired by SNPs, colored by year and annotated with sequence type and SNPs. (b) Maximum-likelihood core-genome phylogenetic tree of all 75 CA *E. coli* isolates annotated by media, MLST, ESBL genes, susceptibility to beta-lactams, beta-lactam exposure, quinolone resistance elements, susceptibility to a quinolone, exposure to quinolones, exposure to immunosuppressant medications, and exposure to chemotherapeutic medications. (c) Bubble plot showing co-occurrence of ESBL genes and quinolone resistance elements, bubbles sized and labeled by the number of isolates, with histograms showing the distribution of the given resistance elements. SNP = single-nucleotide polymorphism; MLST = multi-locus sequence type; ESBL = extended-spectrum beta-lactamase; CiproR = ciprofloxacin resistance; S = susceptible; SDD = susceptible dose dependent; I = intermediate; R = resistant.

in different years, highlighting that these particular lineages have persisted in the community. Isolates within the same SNP cluster carried the same ARGs (Fig. S4).

## Sequence types and antimicrobial resistance in this study

Multi-locus sequence typing identified 23 unique STs. ESBL genes were more common among non-ST131 isolates, although quinolone resistance elements were common throughout (Fig. 3b). Beta-lactam resistance, as determined by antimicrobial suscepti-bility testing (AST), was distributed throughout the sample set, including in isolates

selected only under ciprofloxacin selection and from patients without recent beta-lactam exposure (Fig. 3b). The most common ESBL genes in our collection were $bla_{\text{CTX-M-15}}$ ($n = 14$) and $bla_{\text{EC-15}}$ ($n = 13$), and these genes were those most likely to co-occur with quinolone resistance elements (Fig. 3c). The majority of isolates were resistant to ciprofloxacin ($n = 64$), often in the absence of patient exposure to quinolones, with point mutations in *gyrA*, *parC*, and *parE* more common than resistance gene carriage (Fig. 3b and c). It is worth noting that some patients with ciprofloxacin-resistant *E. coli* and without recent quinolone exposure were recently exposed to chemotherapeutics, immunosuppressants, or both (Fig. 3b), although these medications were not associated with ciprofloxacin resistance ($P = 0.103$ and $0.059$, respectively, Wilcoxon rank-sum test). ESBL genes and quinolone resistance elements frequently co-occurred in this study (Fig. 3c).

Over half of this study's sample set was comprised of ST131 isolates ($n = 39$, 52%), the majority of which encoded the FimH30 gene. We determined that a majority of ST131 isolates encoded quinolone resistance genes and belonged to the H30R sub-lineage ($n = 27$), with a minor fraction also encoding the $bla_{\text{CTX-M-15}}$ gene, confirming their H30Rx status ($n = 6$) (Fig. S5a). We observed that ST131 isolates in this study carried fewer ESBL genes per isolate (Wilcoxon rank-sum test, $P = 0.0025$; Fig. 4a) but more quinolone resistance elements per isolate (Wilcoxon rank-sum test, $P = 0.0108$; Fig. 4b) than non-ST131 isolates. Though ST131 isolates did include co-occurrence of ESBL genes and quinolone resistance elements (Fig. 4c), non-ST131 isolates had a greater diversity of both resistance elements and more frequent co-occurrence (Fig. 4d). Compared to non-ST131 genomes ($n = 36$, 48%), ST131's pangenome encoded for roughly 1,000 additional genes (Fig. S5b).

## Mahmud et al. comparison to ESBL *E. coli* isolates in this study

We leveraged a convenience sample of 87 published ESBL *E. coli* isolates from patient infections at BJH from 2018 to 2019 (Table 2) to compare ARG profiles between clinical isolates and our CA ESBL *E. coli* isolates. A comparison of this study to Mahmud et al. (29) is given in Table 3. Mahmud et al. (29) isolates were derived from the initial culture-positive collection from blood ($n = 35$, 40%) or urine ($n = 53$, 60%) of hospitalized

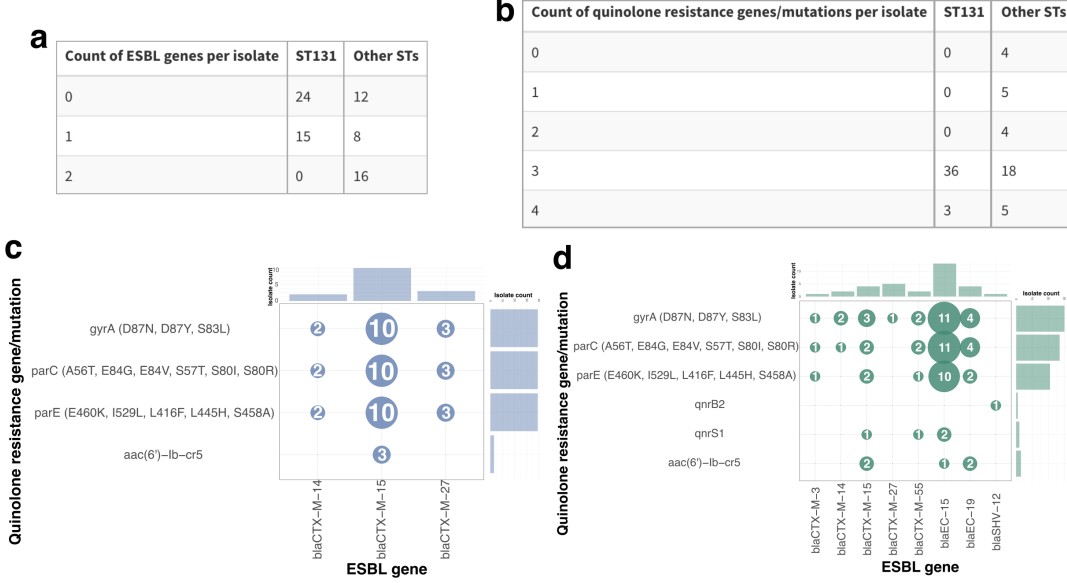

**a**

| Count of ESBL genes per isolate | ST131 | Other STs |
|---|---|---|
| 0 | 24 | 12 |
| 1 | 15 | 8 |
| 2 | 0 | 16 |

**b**

| Count of quinolone resistance genes/mutations per isolate | ST131 | Other STs |
|---|---|---|
| 0 | 0 | 4 |
| 1 | 0 | 5 |
| 2 | 0 | 4 |
| 3 | 36 | 18 |
| 4 | 3 | 5 |

**FIG 4** ESBL genes and quinolone resistance elements of ST131 isolates versus non-ST131 isolates. Tables of the number of isolates with each number of (a) ESBL genes per isolate or (b) quinolone resistance genes per isolate, as determined by AMRFinderPlus. Bubble plots indicating the co-occurrence of ESBL genes and quinolone resistance elements, with bubbles sized and labeled by the number of isolates, in (c) ST131 isolates and (d) non-ST131 isolates. Corresponding histograms show the distribution of the resistance elements. ST = sequence type; ESBL = extended-spectrum beta-lactamase.

**TABLE 2** Patient characteristics in Mahmud et al. (29)

| Variable | Mean (range) or N (%) |
| --- | --- |
| Age, mean (range) | 60 years (20-93 years) |
| Female, N (%) | 49 (56) |
| Medication in last 12 weeks, N (%) | |
| Any antibiotics | 67 (77) |
| Penicillins | 14 (16) |
| Cephalosporins | 45 (52) |
| Carbapenems | 14 (16) |
| Quinolones | 26 (30) |
| Immunosuppressant | 24 (27) |
| Chemotherapy | 23 (26) |

patients. Similar to the 23 CA ESBL *E. coli* from this study, the Mahmud et al. (29) isolates displayed phenotypic susceptibility profiles of ESBL strains, namely, resistance to first-generation cephalosporins, resistance to at least one third-generation cephalosporin, and susceptibility to cephamycins. Phylogenetic analysis identified multiple ST clusters found exclusively within a specific specimen type (Fig. S6a and b). Collectively, this constitutes a diverse sample of 110 ESBL *E. coli* from three distinct specimen types.

We conducted pairwise SNP comparisons between the 23 ESBL *E. coli* isolates from this study and the 87 ESBL *E. coli* isolates from the Mahmud et al. (29) sample set to understand predominant lineages persisting across both sets over time. Using our empirically established $\leq$22 SNP cutoff to identify strain-specific lineages, we identified 10 unique strain networks containing at least two isolates, hereafter referred to as modules. Given the reported plasticity of our ST131 pangenomes and its prevalence in this study, it is unsurprising that the majority of our network modules belonged to ST131 ($n = 8$). Temporal network analysis revealed that most modules (seven of 10) persist across multiple years (Fig. 5a and b). Interestingly, among the three modules recovered within a single calendar year, two belonged exclusively to the sample set of clinical infection isolates (Fig. S6b and c).

## Resistance genes of Mahmud et al. versus CA ESBL *E. coli* in this study

Initial comparisons across isolate source and sample set identified no difference in total ARG carriage (Fig. 5c and d). The carriage of beta-lactam resistance genes was significantly higher in the Mahmud et al. (29) isolates ($P = 0.047$, Wilcoxon rank-sum test; Fig. 5e). When looking specifically at ESBL genes, the overwhelming majority belong to the $bla_{CTX}$ genotype ($n = 19$, 83% for this study, $n = 81$, 92% for Mahmud et al. [29]), followed by the $bla_{TEM}$ genotype ($n = 5$, 22% for this study, $n = 31$, 35% for Mahmud et al. [29]) (Fig. 5f). Both sample sets exhibit highly similar proportions of $bla_{CTX}$ variants, with approximately half belonging to $bla_{CTX-M-15}$ (47% and 49% for this study and Mahmud et al. [29], respectively), followed by the emerging $bla_{CTX-M-27}$ subtype (37% and 28% for this study and Mahmud et al. [29], respectively) (Fig. 5g). Although genomic analysis suggests the potential for persisting lineages across isolate sets, our

**TABLE 3** Comparison of this study and Mahmud et al. (29)

| Variable | This study, N = 102 isolates | Mahmud et al. (29), N = 87 isolates |
| --- | --- | --- |
| Specimen type(s) | Stool | Blood or urine |
| Time period | 2017–2021[a] | 2018–2019 |
| Patient status | Inpatient/observation, outpatient, or ED | Admitted to medical and/or oncology ward |
| Specimen collection timing | <72 h of admission/healthcare encounter | Collected during medical/oncology ward admission or within 24 h of ward admission or discharge |
| Previous healthcare exposures | No inpatient healthcare exposures >24 h in previous 12 weeks | Any |

[a]Aside from one specimen collected in November 2017, the remaining specimens in this study were collected from 2018 to 2021.

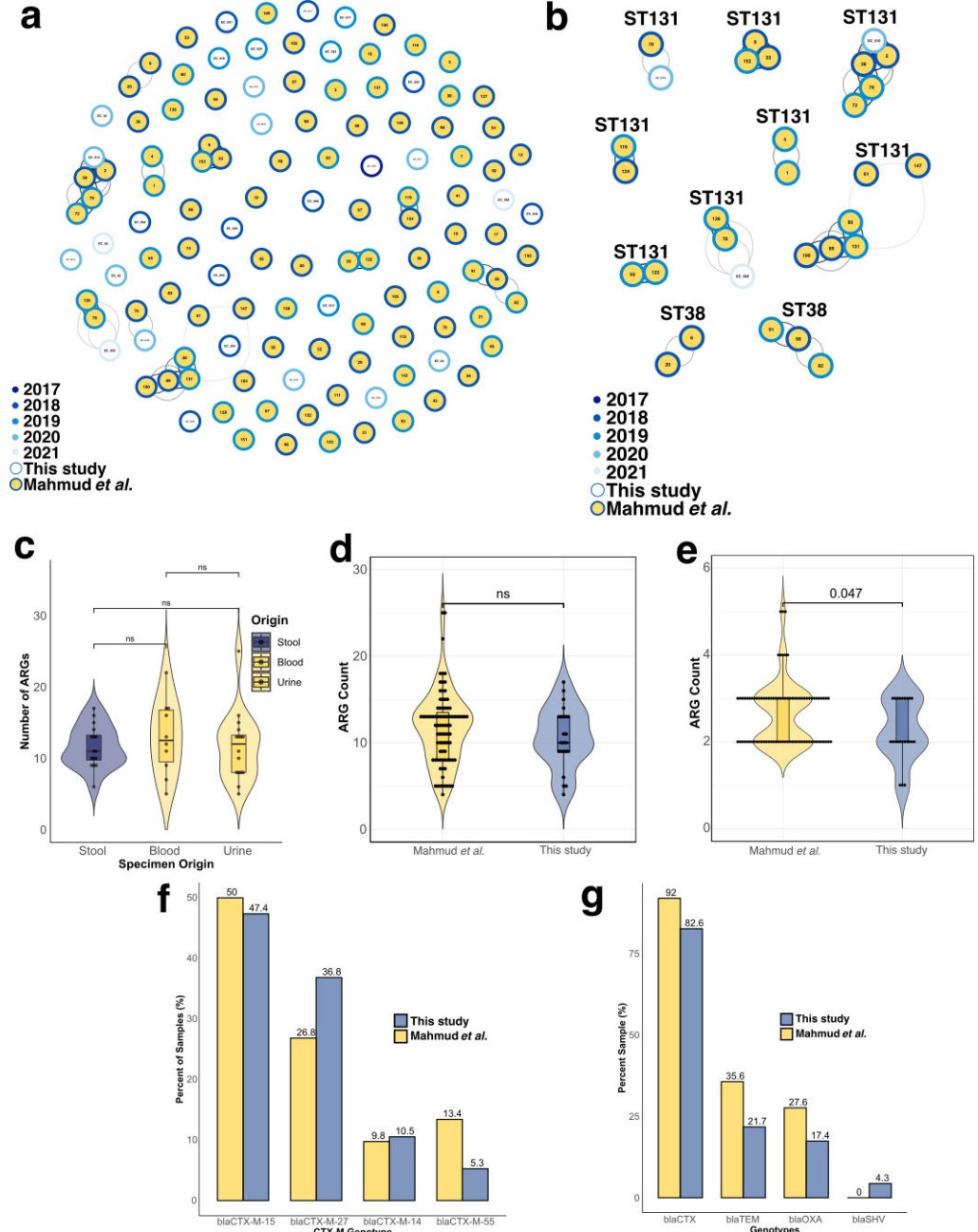

**FIG 5** Genomic surveillance and ARG comparison across isolate sets. (a) Network clustering of 110 CA ESBL *E. coli* (hollow circles) and ESBL *E. coli* from Mahmud et al. infection isolates (filled circles). Each node represents a single isolate, and edges connect highly similar isolate networks colored by sample set. (b) Strain networks of isolates <22 SNPs apart. (c) Total ARG carriage across three isolate sources (stool, blood, and urine). (d) Total ARG carriage across the two sample sets. (e) Beta-lactamase gene frequency across both isolate sets. Dots in (c–e) correspond to individual isolates, and *P*-values were determined by Wilcoxon rank-sum test. (f) Percentage of beta-lactamase genotypes between sample sets. (g) Percentage of CTX-M allelic variants. ESBL = extended-spectrum beta-lactamase; SNP = single-nucleotide polymorphism; ST = sequence type; ARG = antimicrobial resistance gene.

comparative analyses reveal minimal differences in ESBL gene carriage and strikingly similar distributions of $bla_{CTX}$ variants.

## DISCUSSION

Our study describes 75 CA antimicrobial-resistant *E. coli* isolates collected from patients in St. Louis, Missouri, in an effort to better understand the presence of resistance in patients with little recent healthcare exposure. We found abundant ST131, ESBL genes, and quinolone resistance elements in these CA isolates, half of which were derived from patients without recent quinolone or beta-lactam exposure, highlighting that antimicrobial resistance can persist in the absence of recent antimicrobial exposure. Core-genome-based SNP networks support this finding, indicating highly similar resistant *E. coli* recovered from different patients in our sample set. Additionally, this work contextualized our 23 CA ESBL *E. coli* with 87 ESBL *E. coli* from BSIs and UTIs reported by Mahmud et al. (29), finding striking similarity in ARG carriage, *bla* variants, and ESBL gene presence between these two sample sets, again emphasizing the significance of ARGs in CA *E. coli*. Taken together, this work reveals the resistance contribution of CA *E. coli* and demonstrates the importance of monitoring community carriage of antimicrobial-resistant organisms.

ST131 was the most common ST in our samples, which is a notorious ST in healthcare-associated extraintestinal infections (9, 54), and its dominance in our study is consistent with the trend of increasing CA ST131 prevalence (4, 13, 14). Notably, the ST131 isolates in our study carried fewer ESBL genes per genome than non-ST131 isolates, a contradiction with ST131's historic importance as an ESBL carrier (9, 11). These ST131 isolates carried more quinolone resistance elements per isolate than other STs, including in the absence of recent quinolone exposure. It is possible that recent exposures to non-antimicrobial medications, including chemotherapeutics and immunosuppressants, influenced both the gut microbiomes of our patients and the antimicrobial resistance of the *E. coli* specifically (55–58). In an ESBL *E. coli* sample set, it is no surprise that $bla_{CTX-M}$ was the most common ESBL gene, although the dominance of the $bla_{CTX-M-27}$, as opposed to the $bla_{CTX-M-15}$ variant, was unexpected (9, 11), once again underlining the importance of continued resistance monitoring in *E. coli*. Since rapid diagnostic tests such as the BioFire BCID2 only test for CTX-M, the presence of $bla_{EC-15}$ in our sample set emphasizes the need to monitor community-circulating resistance genes to ensure that appropriate tests are being applied (59). Within our cohort of CA *E. coli*, pangenome analysis revealed that the ST131 pangenome included approximately 1,000 more genes than non-ST131 *E. coli*. This expanded pangenome represents a broader repertoire of accessory genes that could contribute to the persistence, adaptability, and success of ST131 across environmental conditions and under varied selective pressures. Notably, our ST131 isolates carried more quinolone resistance elements per isolate compared to other STs. All of our ST131 isolates carried mutations in both gyrase (*gyrA*) and topoisomerase (*parC*, *parE*) genes that determine quinolone resistance, consistent with the increasing prevalence of ciprofloxacin resistance in CA *E. coli* (23). Similarly, all ST1193 isolates carried mutations in both gyrase and topoisomerase, consistent with our media selection and with ST1193 as a lineage (9). This work adds to the reported significance of both ST131 and ST1193 as prevalent quinolone-resistance concerns in the community and highlights the importance of continued evaluation of ST resistance patterns in *E. coli*, particularly in community settings.

Through our core-genome SNP analysis, we found four pairs of isolates separated by fewer than 22 SNPs, suggesting the existence of a common community reservoir from which these patients acquired resistant *E. coli*. It is remarkable that we identified these highly similar isolates from separate patients in a sample set of only 75 isolates across four years. Leveraging this SNP threshold, we observed that more than 15% ($n = 11$) of our patients were colonized with multiple strains of *E. coli* simultaneously. These co-colonization events were revealed under the selective pressures of growth on ESBL Chrome Agar and MacConkey agar with ciprofloxacin, which means that they represent only a subset of drug-resistant microbes from the sampled patients' GI tracts and are a likely underestimate of the extent of *E. coli* strain co-colonization within a single host. Importantly, patients colonized with diverse *E. coli* strains, particularly those with

differing resistance determinants, may serve as important but unidentified reservoirs for continued transmission within healthcare or community settings. This finding suggests that co-colonization and its role in persistence, evolution, and antimicrobial resistance may be underestimated. Our results determined that the majority of co-colonization events occurred within the same ST, which may indicate active within-host divergence. In contrast, co-colonization with *E. coli* of unique STs suggests strain acquisitions from distinct sources or reservoirs. Collectively, these findings underscore the complexity of antibiotic-resistant *E. coli* colonization and resistome dynamics even in the absence of recent antimicrobial exposure.

Comparisons between our 23 CA ESBL *E. coli* in this study and clinical infection-derived ESBL *E. coli* in Mahmud et al. (29) revealed minimal differences in the total ARG carriage. This finding highlights the growing concern that drug-resistant *E. coli* are not confined to the hospital setting but are also well established within the community. Furthermore, when looking exclusively at ESBL gene carriage among the two sample sets, we observed a higher percentage of beta-lactamases in the Mahmud et al. (29) isolates, although we observed little to no difference in the proportion of ESBL geno-types or $bla_{CTX-M}$ variants. Similarities among the proportions of encoded ARGs in CA ESBL *E. coli* and clinical infection ESBL *E. coli* underscore the resistance threat posed by CA organisms, emphasizing the need for continued community-based surveillance to track emerging resistance and potential interactions between community and clinical reservoirs.

Using the prior criterion of ≤22 core-genome SNPs, we identified 10 unique strain networks across both isolate sample sets within the four-year study window. Of these 10 networks, eight belonged to ST131, which further highlights the increasing heteroge-neity among this lineage. Seven of 10 network modules occurred exclusively among Mahmud et al. (29) isolates. Three of these occurred exclusively in UTIs, including a single network module of six isolates which were recovered within a two-year window. We observed three network modules which persisted across both CA isolates in this study and isolates from Mahmud et al. (29), all belonging to ST131. These findings further support the notion that ST131 is a pervasive lineage whose heterogeneity contributes to its global predominance in both the community and clinical settings.

We recognize three key limitations in our study. First, our patient sample set includes stool samples from those with diarrhea, prompting *C. difficile* testing by clinicians, and half of our patients received recent antimicrobials. Therefore, our sample set does not reflect antibiotic-resistant *E. coli* colonization among truly asymptomatic community members, although very few of our patients (*n* = 3) were culture-positive for the presence of *C. difficile*. Nevertheless, we recognize that our use of remnant stool samples from *C. difficile* testing represents a potential bias toward people who may have a more disrupted gut microbiome than healthy, non-hospitalized individuals in the wider community. Secondly, as discussed above, we selected distinct colony morphologies or phenotypic strains during selective culturing, which may not capture the genetic diversity within a single stool specimen. This method inherently ignores colonies that appear morphologically similar or that are not ESBL-producing or resistant to ciprofloxa-cin. Similarly, we recognize that our use of culture-guided selection limited the sensitivity of our *E. coli* detection, which likely contributed to our lack of isolates that could grow on both agars despite genomic evidence of resistance to ciprofloxacin and ESBL. Regard-less, our detection of within-patient co-colonization indicates that future studies should further explore this phenomenon. Finally, we acknowledge our relatively small sample size of 75 isolates and its likely underrepresentation of community-based transmission of *E. coli*. Nevertheless, we were able to identify evidence of CA *E. coli* strain sharing in this study, and we plan for future studies to expand on our findings in larger cohorts that include healthy community members.

Collectively, our study sheds light on resistant *E. coli* in the St. Louis community. We hope this work informs infection treatment practices for CA *E. coli* infections, supports community infection prevention efforts, and ultimately improves patient care.

By presenting evidence of within-patient co-colonization and CA *E. coli* strain sharing, we intend to encourage clinicians and researchers alike to consider diverse *E. coli* acquisition routes. In revealing the strong similarities in ARGs and ESBL genes between CA ESBL *E. coli* in this study and the infection-sourced ESBL *E. coli* from Mahmud et al. (29), we aim to emphasize the resistance threat of CA *E. coli* in our sample. Ultimately, this work serves as additional evidence supporting community-based surveillance for antimicrobial-resistant *E. coli*.

## ACKNOWLEDGMENTS

This work was supported by the Centers for Disease Control and Prevention, Prevention Epicenters grant U54CK000609-05. Its contents are the views of the authors and do not necessarily represent the official views of, nor an endorsement by, CDC/HHS or the U.S. Government. E.E.B. was supported by the National Human Genome Research Institute training grant (T32 HG000045). W.A. received support from the Chancellor's Graduate Fellowship Program at Washington University in St. Louis. The work of Mahmud et al. (29) was supported by an award from the Centers for Disease Control and Prevention (U01CK000587).

We acknowledge the Barnes-Jewish Foundation and the staff of the Edison Family Center for Genome Sciences and Systems Biology and the Genome Technology Access Center. We thank Dr. Kevin Blake, Scientific Editor in the Division of Laboratory and Genomic Medicine, for his contribution in illustrating Fig. 1a and b. Finally, we recognize the participants, without whom this work would not be possible.

## AUTHOR AFFILIATIONS

[1]The Edison Family Center for Genome Sciences and Systems Biology, Washington University School of Medicine, St. Louis, Missouri, USA

[2]Division of Infectious Diseases, Department of Internal Medicine, Washington University School of Medicine, St. Louis, Missouri, USA

[3]Division of Laboratory and Genomic Medicine, Department of Pathology and Immunology, Washington University School of Medicine, St. Louis, Missouri, USA

[4]Department of Biomedical Engineering, Washington University in St. Louis, St. Louis, Missouri, USA

[5]Department of Molecular Microbiology, Washington University School of Medicine, St. Louis, Missouri, USA

[6]Department of Pediatrics, Washington University School of Medicine, St. Louis, Missouri, USA

## AUTHOR ORCIDs

Emily E. Benedict http://orcid.org/0000-0003-3243-3874
Gautam Dantas http://orcid.org/0000-0003-0455-8370
Jennie H. Kwon http://orcid.org/0009-0007-9774-6835

## FUNDING

| Funder | Grant(s) | Author(s) |
| --- | --- | --- |
| Centers for Disease Control and Prevention | U54CK000609-05 | Jennie H. Kwon |
| National Human Genome Research Institute | T32 HG000045 | Emily E. Benedict |
| Centers for Disease Control and Prevention | U01CK000587 | Erik R. Dubberke |

## AUTHOR CONTRIBUTIONS

Emily E. Benedict, Conceptualization, Data curation, Formal analysis, Methodology, Project administration, Validation, Visualization, Writing – original draft, Writing – review and editing | Wesley Agee, Conceptualization, Data curation, Formal analysis, Methodology, Validation, Visualization, Writing – original draft, Writing – review and editing | Tiffany Hink, Data curation, Methodology, Writing – review and editing | Katelyn L. Parrish, Data curation, Methodology, Project administration, Validation, Writing – review and editing | Kimberly A. Reske, Data curation, Methodology, Project administration, Validation, Writing – review and editing | Kate Peacock, Data curation, Methodology, Project administration, Validation, Writing – review and editing | Rachel E. Bosserman, Data curation, Writing – review and editing | Alyssa Valencia, Data curation, Writing – review and editing | Akshay Saluja, Data curation, Writing – review and editing | Elianora Ovchiyan, Data curation, Writing – review and editing | Olivia Arter, Data curation, Writing – review and editing | Kevin Jolani, Data curation, Writing – review and editing | Erik R. Dubberke, Conceptualization, Data curation, Writing – review and editing | Gautam Dantas, Conceptualization, Investigation, Supervision, Writing – review and editing | Jennie H. Kwon, Conceptualization, Funding acquisition, Investigation, Methodology, Resources, Supervision, Writing – review and editing

## DATA AVAILABILITY

All custom scripts are available at https://doi.org/10.5281/zenodo.16883513. Paired sequencing reads and assembled genomes are accessible under BioProject PRJNA1327969.

## ETHICS APPROVAL

This study was approved by the Washington University Human Research Protection Office (IRB #202006166, HRPO #202006166). As this study made use of remnant clinical samples, a waiver of the requirement of informed consent was obtained.

## ADDITIONAL FILES

The following material is available online.

### Supplemental Material

**Fig. S1 (mSystems01591-25-s0001.pdf).** SNP analysis histograms.
**Fig. S2 (mSystems01591-25-s0002.pdf).** ARG annotations of within-patient co-colonizing isolates.
**Fig. S3 (mSystems01591-25-s0003.pdf).** ARG and AST information for all isolates in the present study.
**Fig. S4 (mSystems01591-25-s0004.pdf).** ARG annotations of cross-patient similar isolates.
**Fig. S5 (mSystems01591-25-s0005.pdf).** ST131 accessory genome information.
**Fig. S6 (mSystems01591-25-s0006.pdf).** Network comparison of the Mahmud et al. comparison isolates with the present study isolates.
**Table S1 (mSystems01591-25-s0007.docx).** Present patient and Mahmud et al. patient descriptions.
**Table S2 (mSystems01591-25-s0008.xlsx).** Isolate-associated metadata.
**Table S3 (mSystems01591-25-s0009.xlsx).** Isolate-level genomic data.
**Table S4 (mSystems01591-25-s0010.xlsx).** Isolate-level ARG information.

### Open Peer Review

**PEER REVIEW HISTORY (review-history.pdf).** An accounting of the reviewer comments and feedback.

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
