## [Reviewer comments · mSystems]

Community-associated quinolone-resistant and extended-spectrum beta-lactamase producing *Escherichia coli* are similar to clinical infection isolates by sequence type and resistome

Emily Benedict, Wesley Agee, Tiffany Hink, Katelyn Parrish, Kimberly Reske, Kate Peacock, Rachel Bosserman, Alyssa Valencia, Akshay Saluja, Elianora Ovichyan, Olivia Arter, Kevin Jolani, Erik Dubberke, Gautam Dantas, and Jennie Kwon

Corresponding Author(s): Emily Benedict, Washington University in St Louis School of Medicine

Review Timeline:

Submission Date:	November 11, 2025
Editorial Decision:	December 1, 2025
Revision Received:	December 2, 2025
Accepted:	December 5, 2025

Editor: Zoe Dyson

Reviewer(s): Disclosure of reviewer identity is with reference to reviewer comments included in decision letter(s). The following individuals involved in review of your submission have agreed to reveal their identity: Sadia Isfat Ara Rahman (Reviewer #1)

Transaction Report:

DOI: <https://doi.org/10.1128/msystems.01591-25>

Re: mSystems01591-25 (**Community-associated quinolone-resistant and extended-spectrum beta-lactamase producing *Escherichia coli* are similar to clinical infection isolates by sequence type and resistome**)

Dear Dr. Emily E Benedict:

The accession number provided in the data sharing statement (PRJNA1327969) does not work in either GenBank or ENA. Please review to ensure it is correct and/or arrange for data release as appropriate.

Revision Guidelines

Sincerely,
Zoe Dyson
Editor
mSystems

Reviewer #1 (Comments for the Author):

1. Ensure consistent use of abbreviations (e.g., spell out ESBL at first mention in the abstract).
2. Provide more detail on patient inclusion/exclusion criteria (e.g., how "no inpatient healthcare exposure >24 hours in the previous 12 weeks" was verified).

3. Consider adding a supplementary table summarizing ARGs by sequence type for easier reader reference.

Reviewer #2 (Comments for the Author):

I am satisfied by the changes the authors have made to the manuscript.

Reviewer #3 (Comments for the Author):

Having reviewed the revised manuscript, I thank the authors for their efforts in amending their work and I have no further comments to make.

1. Ensure consistent use of abbreviations (e.g., spell out ESBL at first mention in the abstract).
2. Provide more detail on patient inclusion/exclusion criteria (e.g., how “no inpatient healthcare exposure >24 hours in the previous 12 weeks” was verified).
3. Consider adding a supplementary table summarizing ARGs by sequence type for easier reader reference.

Firstly, these authors would like to thank the editor and all reviewers for their attention and consideration. Below, please find our point-by-point revisions in response to all editor and reviewer comments.

Editor, Dr. Zoe Dyson

Comment to the authors: *“The accession number provided in the data sharing statement (PRJNA1327969) does not work in either GenBank or ENA. Please review to ensure it is correct and/or arrange for data release as appropriate.”*

Author response: We thank Dr. Dyson for her attention in this matter. We have submitted a request to NCBI to release the data by December 16, 2025.

Reviewer #1

Comment to the authors: *“Ensure consistent use of abbreviations (e.g., spell out ESBL at first mention in the abstract).”*

Author response: We thank the reviewer for their reminder, and we have ensured that the abbreviations are indeed spelled out upon first use. This included changes on lines 42, 101, and 166.

Comment to the authors: *“Provide more detail on patient inclusion/exclusion criteria (e.g., how “no inpatient healthcare exposure >24 hours in the previous 12 weeks” was verified).”*

Author response: We appreciate this opportunity for additional clarity, and have added the following language on lines 114-115:

“...for each patient to ensure that all network and non-network healthcare exposures were identified.”

Comment to the authors: *“Consider adding a supplementary table summarizing ARGs by sequence type for easier reader reference.”*

Author response: We appreciate this reviewer’s request for clarity, and have added summary metrics by sequence type to Supplemental Table 4.

Re: mSystems01591-25R1 (**Community-associated quinolone-resistant and extended-spectrum beta-lactamase producing *Escherichia coli* are similar to clinical infection isolates by sequence type and resistome**)

Dear Dr. Emily E Benedict:

Your manuscript has been accepted, and I am forwarding it to the ASM production staff for publication. Your paper will first be checked to make sure all elements meet the technical requirements. ASM staff will contact you if anything needs to be revised before copyediting and production can begin. Otherwise, you will be notified when your proofs are ready to be viewed.

Sincerely,
Zoe Dyson
Editor
mSystems